

# A semi-empirical potential energy surface and line list for H$_2$$^{16}$O extending into the near-ultraviolet

Eamon K. Conway[1,2], Iouli E. Gordon[1], Jonathan Tennyson[2], Oleg L. Polyansky[2], Sergei N. Yurchenko[2], and Kelly Chance[1]

[1]Center for Astrophysics | Harvard and Smithsonian, Atomic and Molecular Physics Division, Cambridge, MA, USA. 02138
[2]Department of Physics and Astronomy, University College London, Gower Street, London WC1E 6BT, United Kingdom

**Correspondence:** Eamon K. Conway (eamon.conway@cfa.harvard.edu)

**Abstract.** Accurate reference spectroscopic information for the water molecule from the microwave to the near-ultraviolet is of paramount importance in atmospheric research. A semi-empirical potential energy surface for the ground electronic state of H$_2$$^{16}$O has been created by refining to almost 4 000 experimentally determined energy levels. These states extend into regions with large values of rotational and vibrational excitation. For all states considered in our refinement procedure, which extend to 37 000 cm$^{-1}$ and $J = 20$, the average root mean squared deviation is approximately 0.05 cm$^{-1}$. This potential energy surface offers significant improvements when compared to recent models by accurately predicting states possessing high values of $J$. This feature will offer significant improvements in calculated line positions for high temperature spectra where transitions between high $J$ states become more prominent.

Combining this potential with the latest dipole moment surface for water vapor, a line list has been calculated which extends reliably to 37 000 cm$^{-1}$. Obtaining reliable results in the ultraviolet is of special importance as it is a challenging spectral region for the water molecule both experimentally and theoretically. Comparisons are made against several experimental sources of cross sections in the near-ultraviolet and discrepancies are observed. In the near-ultraviolet our calculations are in agreement with recent atmospheric retrievals and the upper limit obtained using broad band spectroscopy by Wilson et al. (J. Quant. Spectrosc. Radiat. Transf., 2016, **170**, 194) but do not support recent suggestions of very strong absorption in this region.

## 1 Introduction

Water vapor is a major absorber of light in the terrestrial atmosphere and it interferes with atmospheric retrievals from the microwave to the near-ultraviolet (Lampel et al., 2015). The water molecule dissociates at 41 145.92 cm$^{-1}$ (Boyarkin et al., 2013), and there are almost no rovibrational transitions beyond that. Although the absorption of water vapor in the near-ultraviolet is known to be weak, particularly when compared to features in the infrared, it obscures retrievals of electronic spectra of important (from atmospheric and pollution monitoring perspective) molecules with trace abundances in the terrestrial atmosphere (Fleischmann et al., 2004; Cantrell et al., 1990; Stutz et al., 2000). Retrievals performed in the visible and near-ultraviolet have a long record of success (Gonzalo Gonzalez Abad et al., 2019). Water vapor is one such molecule where retrievals have already been performed in the visible spectrum using the Ozone Monitoring Instrument (OMI) (Levelt et al.,





2018) and accurate results obtained (Wang et al., 2014, 2019). Recent observations indicate water vapor overlaps with near-
ultraviolet absorption features of trace molecules such as $H_2CO$, $O_2$-$O_2$, BrO and HONO (Lampel et al., 2017). The marginal
concentration of these molecules implies that weak water vapor absorption may in fact interfere with their observation.

Satellite missions possessing spectrometers with detection limits extending into the near-ultraviolet are becoming more
popular, for both Earth and planetary studies: Hubble Space Telescope (HST) (NASA), MAVEN (NASA), CUTE (Fleming
et al., 2018), OMI (Levelt et al., 2018) and the recently launched GEMS (Kim et al., 2019) to name but a few. NASA's
TEMPO (*Tropospheric Emissions Monitoring of Pollution*) mission will monitor the air over North and Central America
from 740 to 290 nm and aims to accurately characterize atmospheric pollution (Zoogman et al., 2017). Without accurate
reference spectra through the entire range, this will not be possible. For the principal $H_2{}^{16}O$ isotopologue of water vapor,
the HITRAN2016 (Gordon et al., 2017) database only extends to 400 nm, and while this limit is more than sufficient for
the majority of applications, the increasing demand of remote-sensing missions operating in the ultraviolet suggests that the
HITRAN spectra range needs to be extended to shorter wavelengths.

Computing an accurate line list requires three elements (Lodi and Tennyson, 2010): an accurate potential energy surface
(PES), an accurate dipole moment surface (DMS) and a program capable of solving the nuclear motion problem for the
Schrödinger equation with an exact kinetic energy operator. The recently calculated water line list due to Polyansky et al.
(2018), named 'POKAZATEL', provided the first attempt to model the entire spectrum of water vapor up to dissociation;
POKAZATEL utilized a newly developed PES, the fewer parameter DMS of Lodi et al. (2011) known as LTP2011S and the
DVR3D nuclear motion program (Tennyson et al., 2004). The spectrum predicted by POKAZATEL has been tested against
observations in our own atmosphere and was found to under-absorb in the near-ultraviolet (Lampel et al., 2017). To address this,
a recently developed dipole moment surface (DMS), CKAPTEN (Conway et al., 2018), has been created through extensive
electronic structure calculations and spectra computed with this DMS have been shown to provide improvements over the
POKAZATEL line list for wavelengths down to 400 nm (Conway et al., 2020a).

Semi-empirical adjustments which start from a high quality *ab initio* PES allow energy levels to be calculated to within a
fraction of a wavenumber when compared to experimental measurements (Bubukina et al., 2011; Mizus et al., 2018; Partridge
and Schwenke, 1997; Polyansky et al., 2018). The POKAZATEL PES (note that the POKAZATEL PES and POKAZATEL
line list are distinct entities) extends to dissociation and predicts energy levels with a root-mean square (RMS) error of 0.118
$cm^{-1}$. The uncertainty due to the potential on the calculated transition intensities in the near-ultraviolet is not documented.

The POKAZATEL line list was also designed for high temperature applications (it is complete), yet, as shown below the
POKAZATEL PES only calculates energy levels to high precision for states with low values of total angular momentum $J$.
The PES's accuracy rapidly diminishes as $J$ grows (Polyansky et al., 2018). This rotational effect is not uncommon in semi-
empirical potentials (Bubukina et al., 2011; Mizus et al., 2018; Partridge and Schwenke, 1997). The distribution of rotational
energy levels makes this potential problematic for the generation of high-temperature spectra where transitions between high
$J$ states are important. However, the POKAZATEL line list is complete and includes all transitions involving states up to
$J_{\max} = 72$, where all states with $J \geq 73$ lie above the dissociation threshold.



Recent near-ultraviolet broadband cavity ringdown measurements by Pei et al. (2020) suggest that water vapor may absorb strongly and should have large effects on observations in the 290-350 nm interval. Pei et al. claims that near-ultraviolet water vapor absorption spectra will "significantly affect" the retrievals of ozone and also contribute 0.26 - 0.76 W m$^{-2}$ to the Earth's energy budget. In 2013, the same group performed a similar experiment in the same wavelength region (Du et al., 2013) which also suggested strong absorption in the near-ultraviolet but the two data sets do not agree with each other. While the earlier dataset showed peaks, albeit greatly amplified, at the wavelengths predicted by theory, the second dataset shows no such correlation.

In contrast, Wilson et al. (2016) investigated the absorption of water vapor between 325 - 420 nm and could not replicate the strong absorption features provided by Du et al. (2013). Wilson et al. report an upper bound on the water vapor absorption in this region of $5 \times 10^{-26}$ cm$^2$ molecule$^{-1}$ which is at least a factor of ten lower than the peaks reported by the other studies.

In this work we create a new semi-empirical potential energy surface that accurately models both the rotational behavior of those high $J$ states while also predicting states near dissociation to a reasonable degree of accuracy. With this surface, a new line list that extends into the near-ultraviolet is calculated and used to investigate the available laboratory and atmospheric measurements of water vapor absorption in the blue and near-ultraviolet.

## 2   Method

### 2.1   Fitting the Ab Initio Surface

Approximately 16 000 electronic structure calculations were previously performed for a dipole moment surface at the MR-CI (multi-reference configuration interaction) level of theory utilizing an aug-cc-pCV6Z basis set (Dunning, 1989; Woon and Dunning Jr., 1995; Peterson and Dunning, 2002) and the Douglass-Kroll-Hess Hamiltonian to order two (DKH2) (Conway et al., 2018). These calculations span water bond lengths in the range of 1.3 - 4.0 a$_0$ with angles between 30 - 178°. Setting the energy at the equilibrium configuration, $r_e = 1.8141$ a$_0$ and $\theta_e = 104.52°$, to zero, the maximum energy of these *ab initio* calculations that we consider is 57 423 cm$^{-1}$.

These points need to be fitted to a functional form to obtain an *ab initio* PES; in the fit each data points was weighted as a function of their energy, with weights $w_i$ smoothly reducing towards zero as energy increases. The weighting function considered here is similar to the function used by Partridge and Schwenke (1997). for their 1997 H$_2$$^{16}$O PES. A similar version of this weighting function is also used in an ethylene PES (Delahaye et al., 2014):

$$w_i^{(\text{PES})} = \frac{(\tanh[-\alpha(E_i - V^{\max})] + 1.002002002)}{2.002002002}, \qquad \alpha = 0.006, \quad V^{\max} = 45000 \qquad (1)$$

While constructing the POKAZATEL (Polyansky et al., 2018) potential energy surface, Polyansky et al. found that a single surface could not accurately predict energies from the bottom of the well up to dissociation, hence they follow the procedure of Varandas (1996) and define a piece-wise potential. The same methodology was recently used to creating a PES for the C$_3$ molecule (Rocha and Varandas, 2018). We are also interested in accurately predicting energies that extend into the near-





ultraviolet and so, we too use a piece-wise defined potential as given by

$$V(r_1, r_2, \theta) = V_{\text{low}}(r_1, r_2, \theta) \times \chi^E(r_1, r_2, \theta) + V_{\text{up}} \times (1 - \chi^E(r_1, r_2, \theta)) \tag{2}$$

where $\chi^E$ is a switching function dependent upon energy ($E$):

$$\chi^E(r_1, r_2, \theta) = \frac{1}{2}\left[1 + \tanh\left((V_{\text{up}}(r_1, r_2, \theta) - \zeta_s)\left(\frac{1}{\beta} + \frac{\Delta E^2}{\beta^3}\right)\right)\right] \tag{3}$$

and $r_1$, $r_2$ and $\theta$ are the corresponding values of the bond lengths and inter-bond angle. This function ensures smoothness and the parameters $\zeta_s$ and $\beta$ control the range of the switch. Our values are similar to those of the POKAZATEL PES, except our switching point $\zeta_s$ is different. By lowering our $\zeta_s$ from the 35 000 cm$^{-1}$ value of POKAZATEL to 30 000 cm$^{-1}$, we allow high order parameters in $V_{\text{low}}$ to have greater influence on the upper levels.

Due to the difficulty of fitting data in different energy regions, it is helpful to begin with a well defined functional form, hence the starting point for $V_{\text{up}}$ in our new PES is the $V_{\text{up}}$ function of the POKAZATEL potential. However for $V_{\text{low}}$, we employ a new functional form defined as

$$V_{\text{low}}(r_1, r_2, \theta) = C_{000}\, G(\theta)\, F(r_1, r_2) + \sum_{ijk} C_{ijk}\, \zeta_1^i \zeta_2^j \zeta_3^k\, D(\theta)\, F(r_1, r_2) +$$

$$D_1(1 - e^{-\alpha r_{1e}})^2 + D_1(1 - e^{-\alpha r_{2e}})^2 + D_2 e^{-|r_{12}|} \tag{4}$$

where $r_{ie} = (r_i - r_e)$ for $i = 1, 2$. $r_{12}$ is the separation between the two hydrogen atoms, while $r_e = 1.8141\, a_0$ is the equilibrium bond length and $\theta_e = 104.52^\circ$ is the angle at equilibrium. $\alpha$ was determined from a series of optimizations and the optimal value was found to be 1.24. $D_1$ and $D_2$ were also floated during our initial linear least square fits and are set to $42778.44$ and $683479.329404$ cm$^{-1}$ respectively. The expansion variables $\zeta_1, \zeta_2$ and $\zeta_3$ are defined as

$$\zeta_1 = (r_1 + r_2)/2 - r_e, \; \zeta_2 = (r_1 - r_2)/2, \; \zeta_3 = \cos\theta - \cos\theta_e. \tag{5}$$

$G(\theta)$ and $F(r_1, r_2)$ are dimensionless damping functions that constrain the potential in the limits of $\theta \to 0$ and $r_{1,2} \to \infty$. These are defined as:

$$G(\theta) = \tanh\frac{\left(20(\frac{\theta}{\theta_e}) - 3.002002002\right)}{2.002002002} + 0.5$$

$$F(r_1, r_2) = (0.999821745456)e^{-0.81(r_{1e}^2 + r_{1e}^2)}. \tag{6}$$

The number of parameters $C_{ijk}$ were optimized to provide the lowest RMS deviation from the underlying *ab initio* data such that there are also no 'holes' created from over-fitting. A 'hole' is an unphysical feature of a PES that often appears as a continuous (although not always) drop/dip in the surface, where it should instead be smooth. We found that using 250 parameters provided the lowest RMS deviation of 35 cm$^{-1}$ from the electronic structure calculations. This value is large due to the large discrepancy between our *ab initio* data points and $V_{\text{up}}$ from POKAZATEL rather than from our fitting of $V_{\text{low}}$. The 250 parameters used here is close to the 241 parameters taken by Bubukina et al. (2011) and Mizus et al. (2018), and the 245





of Partridge and Schwenke (1997). The maximum values of $i, j, k$ that we consider are 10, 8 and 15 respectively. In addition to the fitted *ab initio* surface, we also include a QED correction to our *ab initio* PES via the one-electron Lamb shift (Pyykkö et al., 2001) and a second order relativistic energy correction (Quiney et al., 2001).

For quanta in $\nu_1$ and $\nu_3$, i.e the stretching modes, Schwenke (2001) discovered that his Born-Oppenheimer diagonal corrections (BODC), also known as the adiabatic correction, did not agree with those calculated by Zobov et al. (1996). The two calculations did however exhibit better agreement for the different quanta of bend in $\nu_2$. The adiabatic correction is known to be large for high stretch modes (Polyansky et al., 2013), particularly for those in the visible and near-ultraviolet which we are interested in. However, neither source is well tested nor suited for such energetic states, hence we chose to omit this correction

to our surface and rely on fitting to experiment to incorporate this effect.

   The non-adiabatic correction is an important contribution to any high-accuracy potential (Partridge and Schwenke, 1997; Schwenke, 2001; Bubukina et al., 2011; Mizus et al., 2018; Polyansky et al., 2013). For high-temperature spectra, transitions involving high values of the total angular momentum, $J$, become significantly more prominent and, as the non-adiabatic correction grows approximately as $J^2$ (Bunker and Moss, 1980), non-adiabatic effects are more important. For this reason, we

follow Bubukina et al. (2011) and embed these corrections within our Hamiltonian as new kinetic energy operators which are functions of operators $\hat{J}_{XX}$, $\hat{J}_{YY}$ and $\hat{J}_{ZZ}$. The coefficients before these operators are the values determined from Schwenke (2001) multiplied by a factor of 1.1, which he suggests, times optimized values from Bubukina et al. In total, this gives (in a.u):

$$(6.48156 \times 10^{-10})\hat{J}_{XX}$$
$$(4.86799 \times 10^{-10})\hat{J}_{YY}$$
$$(3.94597 \times 10^{-10})\hat{J}_{ZZ} \tag{7}$$

### 2.2   Nuclear Motion Calculations

We use the DVR3D (Tennyson et al., 2004) suite of programs for solving the nuclear motion problem. For these calculations we take Radau coordinates with a bisector embedding and use a 55 by 40 discrete variable representation (DVR) grid with Morse oscillator like functions in $r$ and associated Legendre polynomials in $\theta$, respectively. The DVR for these basis sets is

140 constructed using Gaussian quadrature schemes in associated-Laguerre and associated-Legendre polynomials respectively in $r$ and $\theta$. For the Morse oscillator-like functions, we take $r_E = 3.0$, $\omega = 0.007$ and $\beta = 0.25$ (all in a.u.), which are the values used to compute the POKAZATEL line list. For the vibrational problem, matrices of dimension 3 500 are diagonalized and used as a basis for the full rovibrational problem. For this, matrices of dimension $600(J + 1 - p)$ are diagonalized, where $J$ is the total angular momentum and $p$ is the parity ($p = 0$ or 1). Nuclear masses have been used throughout.

These parameters have been optimized for the initial $J = 0$ problem such that vibration energies below 27 000 cm$^{-1}$ are well converged to better than 0.01 cm$^{-1}$, while for energies at 37 000 cm$^{-1}$ the convergence error is less than 0.03 cm$^{-1}$.





Table 1: Average residuals of energy levels calculated with this PES, the POKAZATEL PES (Polyansky et al., 2018) and the PES15K PES (Mizus et al., 2018) compared to MARVEL levels (Császár et al., 2007; Furtenbacher and Császár, 2012; Tennyson et al., 2014) with $J \leq 20$.

| PES | This Work | | POKAZATEL | | PES15K | |
|---|---|---|---|---|---|---|
| $J$ | $N_J$ | $\sigma$ (cm$^{-1}$) | $N_J$ | $\sigma$ (cm$^{-1}$) | $N_J$ | $\sigma$ (cm$^{-1}$) |
| E $\leq$ 15 000 cm$^{-1}$ | | | | | | |
| 0 | 50 | 0.03946 | 50 | 0.02923 | 50 | 0.01523 |
| 1 | 149 | 0.02973 | 151 | 0.02539 | 151 | 0.01084 |
| 2 | 258 | 0.02956 | 256 | 0.02118 | 257 | 0.01054 |
| 3 | 360 | 0.02843 | 363 | 0.01828 | 365 | 0.01126 |
| 4 | 461 | 0.02703 | 460 | 0.01640 | 461 | 0.00705 |
| 5 | 564 | 0.02665 | 566 | 0.02410 | 563 | 0.00831 |
| 6 | 641 | 0.02651 | 640 | 0.03322 | 639 | 0.01049 |
| 7 | 700 | 0.02872 | 703 | 0.04919 | 698 | 0.01473 |
| 8 | 729 | 0.02697 | 730 | 0.06102 | 729 | 0.01820 |
| 9 | 735 | 0.02912 | 737 | 0.07750 | 736 | 0.03304 |
| 10 | 707 | 0.03212 | 704 | 0.08884 | 708 | 0.02740 |
| 11 | 662 | 0.03254 | 661 | 0.09984 | 664 | 0.03033 |
| 12 | 641 | 0.03651 | 632 | 0.10682 | 640 | 0.03185 |
| 13 | 603 | 0.03847 | 589 | 0.11009 | 603 | 0.03735 |
| 14 | 554 | 0.03894 | 535 | 0.11618 | 557 | 0.04162 |
| 15 | 502 | 0.03957 | 473 | 0.12102 | 508 | 0.04691 |
| 16 | 480 | 0.03931 | 433 | 0.11932 | 482 | 0.04944 |
| 17 | 464 | 0.04322 | 414 | 0.12201 | 461 | 0.05870 |
| 18 | 439 | 0.04474 | 390 | 0.12950 | 438 | 0.06495 |
| 19 | 425 | 0.04872 | 363 | 0.12984 | 421 | 0.06882 |
| 20 | 402 | 0.05577 | 338 | 0.12996 | 399 | 0.07676 |
| Total | 10526 | 0.03453 | 10191 | 0.08047 | 10530 | 0.03160 |
| E $\leq$ 26 000 cm$^{-1}$ | | | | | | |
| 0 | 82 | 0.04560 | 82 | 0.03184 | - | - |
| 1 | 273 | 0.03683 | 276 | 0.02976 | - | - |

( To be continued)





| PES | This Work | | POKAZATEL | | PES15K | |
| :---: | :---: | :---: | :---: | :---: | :---: | :---: |
| $J$ | $N_J$ | $\sigma$ (cm$^{-1}$) | $N_J$ | $\sigma$ (cm$^{-1}$) | $N_J$ | $\sigma$ (cm$^{-1}$) |
| 2 | 485 | 0.03799 | 485 | 0.02704 | - | - |
| 3 | 679 | 0.03949 | 683 | 0.02594 | - | - |
| 4 | 832 | 0.03583 | 849 | 0.02888 | - | - |
| 5 | 993 | 0.03511 | 1007 | 0.03557 | - | - |
| 6 | 1061 | 0.03574 | 1074 | 0.04327 | - | - |
| 7 | 1105 | 0.03687 | 1123 | 0.06028 | - | - |
| 8 | 1049 | 0.03376 | 1049 | 0.07379 | - | - |
| 9 | 981 | 0.03447 | 981 | 0.08988 | - | - |
| 10 | 880 | 0.03580 | 870 | 0.10130 | - | - |
| 11 | 772 | 0.03340 | 765 | 0.11047 | - | - |
| 12 | 703 | 0.03821 | 688 | 0.11478 | - | - |
| 13 | 643 | 0.03940 | 621 | 0.11491 | - | - |
| 14 | 599 | 0.04112 | 568 | 0.12002 | - | - |
| 15 | 540 | 0.04059 | 488 | 0.12256 | - | - |
| 16 | 508 | 0.04026 | 447 | 0.12119 | - | - |
| 17 | 484 | 0.04403 | 420 | 0.12342 | - | - |
| 18 | 469 | 0.04581 | 400 | 0.13151 | - | - |
| 19 | 456 | 0.05064 | 375 | 0.13184 | - | - |
| 20 | 436 | 0.05621 | 431 | 0.13117 | - | - |
| Total | 14030 | 0.03833 | 13682 | 0.07989 | | |
| $E \leq 37\,000$ cm$^{-1}$ | | | | | | |
| 0 | 98 | 0.15349 | 96 | 0.08515 | - | - |
| 1 | 312 | 0.10341 | 315 | 0.08023 | - | - |
| 2 | 564 | 0.12777 | 572 | 0.10604 | - | - |
| 3 | 727 | 0.07496 | 738 | 0.06679 | - | - |
| 4 | 873 | 0.07327 | 890 | 0.05624 | - | - |
| 5 | 1032 | 0.06786 | 1052 | 0.06131 | - | - |
| 6 | 1073 | 0.04407 | 1088 | 0.05067 | - | - |
| 7 | 1113 | 0.04561 | 1131 | 0.06486 | - | - |
| 8 | 1051 | 0.03535 | 1060 | 0.07391 | - | - |
| Total | 6843 | 0.06475 | 6942 | 0.06695 | | |



### 2.3 Creating a Semi-Empirical PES

PES refinement is a technique where one adjusts the underlying *ab initio* surface to reproduce measured data to a high degree of
accuracy, often to within a fraction of a wavenumber (Huang et al., 2012; Polyansky et al., 2018; Mizus et al., 2018; Bubukina
et al., 2011). The method of Yurchenko et al. (2003) has been successfully applied to numerous $H_2O$ potentials (Polyansky
et al., 2018; Mizus et al., 2018; Bubukina et al., 2011), as well as to TiO (McKemmish et al., 2019), $AsH_3$ (Coles et al., 2019),
$NH_3$ (Coles et al., 2018), $CH_3Cl$ (Owens et al., 2018) and $C_2H_4$ (Mant et al., 2018). In this procedure, one maintains the
overall structure of the underlying *ab initio* surface while simultaneously optimizing the parameters of the fit. This prevents
the development of unwanted 'holes' while refining.

Overall, we are trying to minimize:

$$X = \sum_i (\Delta_i^{(\mathrm{obs})})^2 w_i^{(\mathrm{obs})} + f \sum_j (\Delta_j^{(\mathrm{ai})})^2 w_j^{(\mathrm{ai})} \tag{8}$$

where $\Delta_i^{(\mathrm{obs})}$ is the typical observed minus calculated DVR3D ro-vibrational energy and similarly $\Delta_j^{(\mathrm{ai})}$ is the difference
between *ab initio* and calculated potential energies. The factor $f$ is the 'weight' of our semi-empirical PES to our initial *ab
initio* surface. Setting $f$ too large can result in over-fitting if the sum over $j$ and/or $i$ is too small.

The Hellman-Feynmann theorem allows us to efficiently calculate the derivative of an energy level with respect to a particular
parameter in our potential, required for the least-squares fit. With this, we can iterate and optimize the parameters of the PES
to reduce the deviation of our semi-empirical energies from the observed levels. The MARVEL (measured active rotational-
vibrational energy levels) procedure (Furtenbacher et al., 2007; Császár et al., 2007; Furtenbacher and Császár, 2012) was
originally constructed for a IUPAC study of water spectra (Tennyson et al., 2014). The resulting empirical energy levels
for $H_2^{16}O$ (Tennyson et al., 2013) have been subsequently been updated in response to both improvements to the MARVEL
algorithm (Tóbiás et al., 2019) and to the availability of new data (Furtenbacher et al., 2020). We refine our potential to updated
MARVEL energy levels with $J = 0, 2, 5, 10, 15$ and $20$, representing approximately 4 000 states. The more recent potentials
for water vapor (Shirin et al., 2003; Polyansky et al., 2018; Mizus et al., 2018; Bubukina et al., 2011) have been limited to
refinement of states with $J = 0, 2$ and $5$, which is not sufficient to accurately predict high $J$ levels.

The only near-ultraviolet energy levels available for $H_2^{16}O$ come from the multiphoton experiments of Grechko et al. (2010,
2009) and span states below $J \cong 7$. The reduced number of measurements in the blue-violet and near-ultraviolet makes the $V_{\mathrm{up}}$
particularly difficult to refine accurately. More high resolution experimental work in these regions would be welcome.





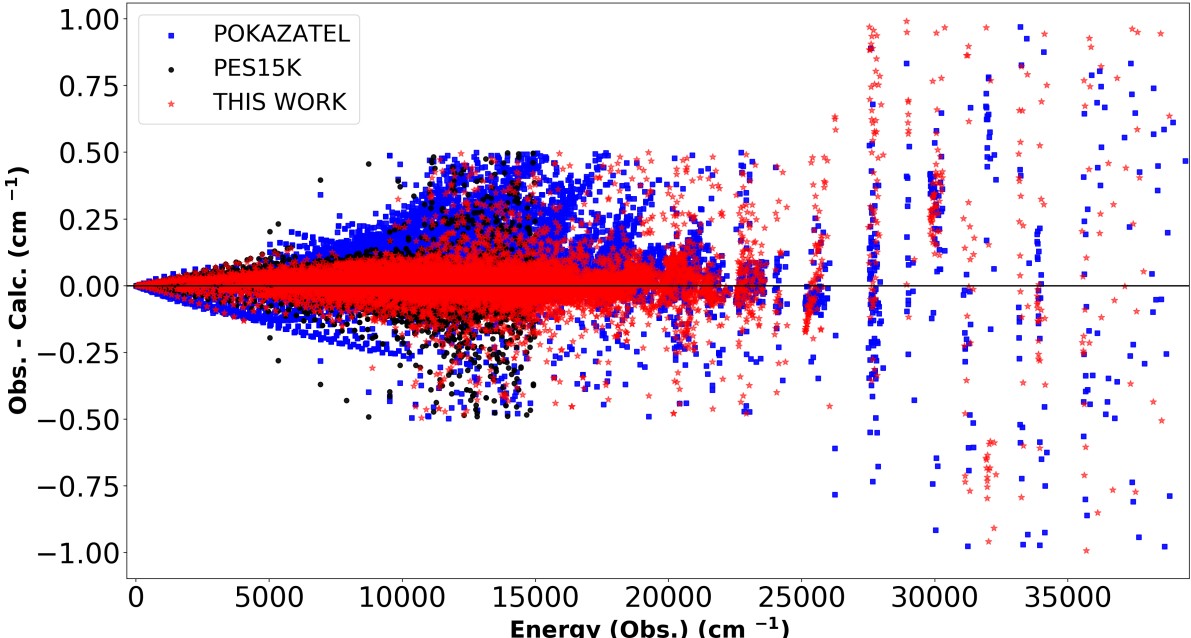

**Figure 1.** Calculated energy levels obtained from the POKAZATEL (Polyansky et al., 2018) surface, PES15K (Mizus et al., 2018) surface and this work compared to those in the MARVEL (Császár et al., 2007; Furtenbacher and Császár, 2012; Tennyson et al., 2014) database

## 3 Results

### 3.1 PES Refinement

For our initial un-refined *ab initio* PES, the average deviation from the MARVEL $J = 0$ *ab initio* vibration band origins (VBOs) below $37\,000$ cm$^{-1}$ is approximately $2$ cm$^{-1}$, a figure dominated by overtones in $\nu_2$. Refining to the VBOs alone is known not to produce accurate results (Schryber et al., 1997). However, fits to $J = 0$ levels are significantly faster and provides a good starting point for refining using non-zero $J$ states.

For the first refinement of $J = 0$ VBOs, we set the weight of all levels with energies greater than $26\,000$ cm$^{-1}$ to 0.1, while those less than this carry a weight of 1. This ratio of 10:1 was chosen such that we can include all states in the refinement without deteriorating the residuals of the lower states. The weight of our semi-empirical PES to the underlying *ab initio* surface was fixed at $1\,000$, which is large enough to provide accurate results, while also small enough to prevent the formation of undesirable 'holes'. For this process, $V_{\mathrm{up}}$ was held constant. Doing this allowed us to reduce our average RMS error from the MARVEL VBOs to only $0.08$ cm$^{-1}$.



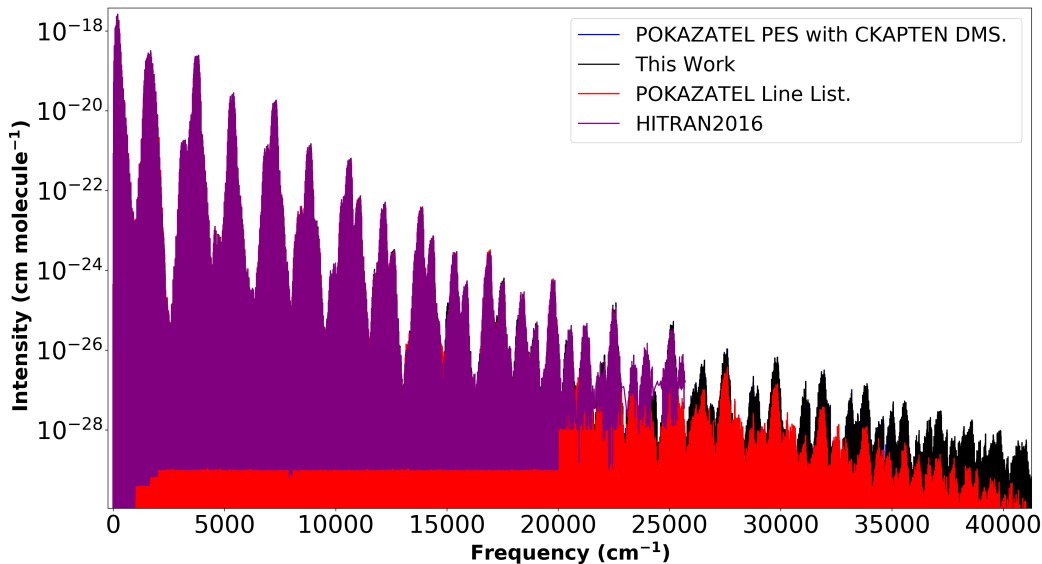

**Figure 2.** Transition intensities from the POKAZATEL line list (Polyansky et al., 2018) this work representing our new PES with the CKAPTEN DMS (Conway et al., 2018), the POKAZATEL PES combined with the CKAPTEN DMS, and HITRAN2016 (Gordon et al., 2017)

For the second step, the ratio of weights for those states below 26 000 cm$^{-1}$ to those above this limit are now switched compared to the previous refinement of $V_{\text{low}}$. 61 of the lowest order parameters in $V_{\text{up}}$ are optimized to improve the agreement between both our *ab initio* data points and the MARVEL levels, while $V_{\text{low}}$ was held fixed. For this refinement of $V_{\text{up}}$, $f$ carries the same value as the previous step and is 1 000.

For the third stage, we return to $V_{\text{low}}$ and focus on the refinement of energies in higher $J$ states, notably $J = 2,\ 5,\ 10,\ 15$ and 20. The weighting criteria remains the same as in step one and $V_{\text{up}}$ was not optimized. Next, for step four, we apply the weighting criteria of step two and refine $V_{\text{up}}$ to states in $J = 0,\ 2,\ 5,\ 10,\ 15$ and 20 and hold $V_{\text{low}}$ fixed. Although there are no known near-ultraviolet states with $J = 10,\ 15$ and 20, the low order parameters in $V_{\text{up}}$ potentially interact very weakly with the lower states and it is important to include these in the optimization such that we do not lose the rotational dependence of

these levels. This step is repeated several more times, each time gradually increasing $f$ towards $10^5$. Increasing $f$ above this provided no improvement in the RMS error and this concluded the refinement of $V_{\text{up}}$.

For the final optimization of our potential, we refine $V_{\text{low}}$ to states in $J = 0,\ 2,\ 5,\ 10,\ 15$ and 20 using the 10:1 ratios of step one while also gradually increasing $f$ to $10^{10}$. Going beyond this offered no improvement in the final RMS error and only increases the risk of over-refining. This $f$ value is significantly larger than that used in the final refinement of $V_{\text{up}}$, which is

entirely justified by there being significantly fewer states in the near-ultraviolet.





It is common to provide a breakdown of residuals for the VBOs in a long table; however, as already described, these states alone cannot be used to measure how well a potential can calculate energy levels. Hence, in Table 1, we provide the average deviation of the calculated energy levels using our new potential, the POKAZATEL potential and the PES15K potential to those MARVEL states with $J \leq 20$. Firstly, we must acknowledge that PES15K is excellent at reproducing those energy levels below 15 000 cm$^{-1}$ with $J \leq 9$, but above this $J$ threshold, the residuals begin to increase and eventually surpass ours. A similar situation occurs for the POKAZATEL surface, but the RMS error now increases much more rapidly with $J$. This is most likely due to these potentials only being refined to states in $J = 0$, 2 and 5. Our new potential offers lower residuals for those high $J$ states while also providing relatively accurate energies into the near-ultraviolet. For high values of $J$, it is also worth noting that, of the three potential surfaces, there are significantly fewer calculated levels from the POKAZATEL PES matched with those in MARVEL despite the same matching criteria being used for all. For the purpose of reproducability, we provide a VBO comparison in the supplementary material.

Figure 1 plots the residuals used to construct Table 1, which compares the calculated energies to those from MARVEL using our new potential, the POKAZATEL potential and PES15K surface. The rotational dependence of the POKAZATEL PES is clear. The Fortran F90 subroutine for our new semi-empirical PES, which we call 'HOT_WAT' is provided in the supplementary material.

### 3.2 Calculation of an ultraviolet line list

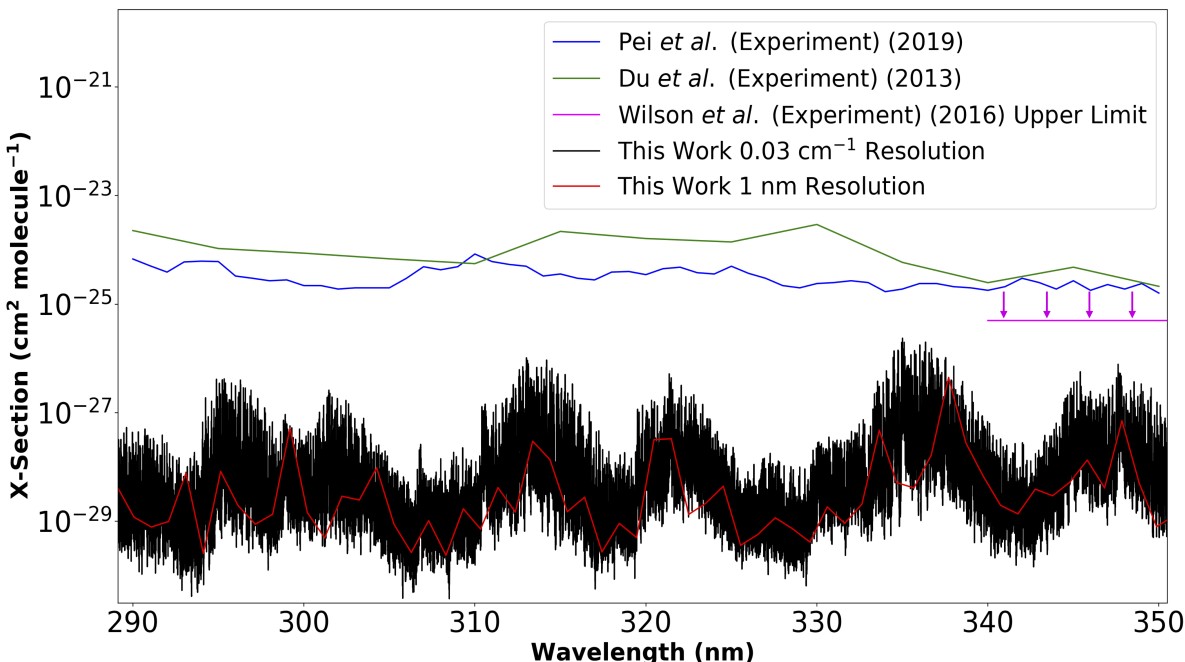

**Figure 3.** Cross sections calculated using our new PES with the CKAPTEN DMS (Conway et al., 2018) at two different resolutions, compared to the measurements of Du et al. (2013), Pei et al. (2020) and the upper limit of Wilson et al. (2016).





To generate transition intensities, we require an accurate dipole moment surface. The CKAPTEN (Conway et al., 2018) surface has previously been shown to provide reliable dipole values (Conway et al., 2020a) and hence, we will use this DMS to calculate our spectra. We compute a line list for $H_2^{16}O$ that extends to 41 200 cm$^{-1}$, i.e. beyond the shortest wavelength that will be accessible by the NASA TEMPO mission which is 290 nm (Zoogman et al., 2017). The accuracy of this line list is not verified for transitions with frequencies beyond 37 000 cm$^{-1}$ and this region may be susceptible to basis set convergence issues. In HITRAN (Gordon et al., 2017) units, the minimum intensity considered here is $10^{-32}$ cm molecule$^{-1}$ and $J_{max} = 20$, all assuming 296 K. There are no transitions in the near-ultraviolet that include $J = 20$ which have intensities surpassing our $10^{-32}$ cm molecule$^{-1}$ threshold. We then proceed to 'MARVELize' this line list, meaning, we replace, where possible, our calculated energy levels with empirical ones from MARVEL, which also allows us to add extra quantum labels ($K_a$, $K_c$, $\nu_1$, $\nu_2$, $\nu_3$) on top of the rigorous labels $J$, parity and symmetry. This process is described in more detail in (Conway et al., 2020a).

In an earlier study (Conway et al., 2018), we generated near-ultraviolet spectra with the POKAZATEL potential and CKAPTEN DMS, although the thresholds used were different to those used here. The maximum transition frequency considered in the previous study was 35 000 cm$^{-1}$ with $J_{max} = 14$ and the minimum intensity considered was $10^{-30}$ cm molecule$^{-1}$, but these criteria should be sufficient for comparison studies in the near-ultraviolet. Comparing these calculations to our new ones will allow us to ascertain how different potential surfaces influence intensities.

In Figure 2, we plot transition intensities from our new calculations, the POKAZATEL line list, HITRAN2016 and our old calculations previously described. For transitions in the IR, the line lists show little deviation, however, as transitions extend further into the blue, differences become significantly more pronounced and in general, the POKAZATEL intensities appear too weak. At 19 000 cm$^{-1}$, the first absorption feature not well represented by the POKAZATEL line list appears. For wavelengths extending from 500 nm to 400 nm, transition intensities in the HITRAN2016 $H_2^{16}O$ line list are of comparable magnitude to ours and are in general, made up from previously published theoretical models, notably BT2 (Barber et al., 2006) and Lodi et al. (2011) data. Atmospheric observations by Lampel et al. (2017) suggest HITEMP2010 (Rothman et al., 2010) (mostly BT2 data) predicts absorption features of water vapor in the visible more accurately than the POKAZATEL line list; hence, it is reasonable to assume POKAZATEL also under-absorbs at 19 000 cm$^{-1}$. At the 400 nm limit of HITRAN2016, we begin to notice larger differences in the intensities, although our new data agrees much better with POKAZATEL.

Comparing our new line list to the old calculations indicates that the new potential does not greatly alter the intensities, which was expected as for stable transitions the DMS controls the magnitude of the absorption (Lodi and Tennyson, 2012). Hence, the differences which are observed in the near-ultraviolet are due to differences in the underlying dipole surfaces. The POKAZATEL line list was computed with the LTP2011S surface of Lodi et al. (2011), where 'S' signifies that this surface is a fewer parameter fit to their *ab initio* dipoles and is therefore more stable in energetic regions.

Lampel et al. (2017) evaluated this POKAZATEL line list in the near-ultraviolet and comments that the feature at approximately 363 nm is underestimated by factor of 2.4±0.7, where the largest contribution to this uncertainty is from the observation. In Figure 2, there is a visible drop in the calculated POKAZATEL cross sections that begin just beyond 25 000 cm$^{-1}$. To verify that our new line list correctly models this feature, we sum transition intensities in both line lists that are within 27 000 cm$^{-1}$





– 27 800 cm$^{-1}$. The ratio of our summed intensities to POKAZATEL is 3.08, which is in within the uncertainty of Lampel et al. Despite this improvement, further validation is required to verify the entire line list. Future work is planned for this.

In 2013, Du et al. (2013) report measurements of a strong, broadband near-ultraviolet absorption spectrum of water in the 350-290 nm region; these absorptions could not be detected by Wilson et al. (2016). The instrumental setup used by Wilson

et al. enabled them to place an upper limit of absorption in this region of $5 \times 10^{-26}$ cm$^2$ molecule$^{-1}$. More recently, Pei et al. (2020) made new measurements in the same region. In order to generate cross sections, we apply approximate air-broadening coefficients ($\gamma_{air}$) which are computed as functions of $J'$ and $J''$ (Rothman et al., 2010) to our new line list and calculate cross sections using the HITRAN API (HAPI) code (Kochanov et al., 2016) at resolutions of 0.03 cm$^{-1}$ and 1 nm with the Voigt profile. It is important to note that the cross-sections reported by Pei et al. are in 1 nm step sizes. Figure 3 compares our

calculations to each of these data sets. The new measurements of Pei et al. give cross sections of comparable magnitude to those of Du et al. but do not resemble any feature in our line list. Importantly our calculated cross sections do not exceed the upper limit of Wilson et al. that extends to 340 nm.

Both Pei et al. and Du et al. suggest that water vapor absorption in the 290 – 350 nm window should be of the order of $10^{-24}$ cm molecule$^{-1}$, which is of comparable magnitude to features observed at 20 000 - 22 750 cm$^{-1}$ (see Figure 2) (500

- 450 nm). Pei et al. suggest this increased water vapor absorption is due to an absorption band between different electronic states, however, the nearest electronic state is an unbound $^1B_1$ state which corresponds to the spectral feature at approximately 170 nm as confirmed by numerous experiments (Chung et al., 2001; Mota et al., 2005; Cantrell et al., 1997b, a). These experiments show that absorption decreases exponentially with increase of the wavelength (i.e. decrease of the wavenumber), as expected considering the upper state is unbound. In order for these electronic transitions to absorb more in the red one needs

to populate high vibrational levels of the ground state, which is not possible at atmospheric temperatures. At room temperature, this band is unlikely to affect absorption in this 290 - 350 nm interval to the degree quoted by Pei et al. Conversely our line list, which predicts greatly reduced cross sections in this reason appear to be in line with atmospheric observations. We are currently collaborating with atmospheric scientists at the Center for Astrophysics | Harvard and Smithsonian (Wang et al., 2014, 2019; Gonzalo Gonzalez Abad et al., 2019) to further investigate this near-ultraviolet absorption by water vapor but

this effort would greatly benefit from further experimental research. Initial tests will focus on data obtained from the Ozone Monitoring Instrument (OMI) (Levelt et al., 2018).

Our calculated line list is available in the supplementary material and assumes 100% H$_2$$^{16}$O isotopic abundance.

## 4 Conclusions

A new semi-empirical potential energy surface for the main water vapor isotopologue is created by refining (Yurchenko et al.,

2003) the *ab initio* model to approximately 4 000 MARVEL (Császár et al., 2007; Furtenbacher and Császár, 2012) energy levels. These states extend to 37 000 cm$^{-1}$ and are possess total angular momenta values of $J = 0, 2, 5, 10, 15$ and 20. By considering such a large range of total angular momenta, we manage to accurately recover the rotational behavior of the energy levels. Comparisons made against the most recent semi-empirical potential energy surfaces (PESs) for water vapor (Mizus





et al., 2018; Polyansky et al., 2018) show our new surface provides lower residuals. For energy levels in $J = 20$, our new
surface predicts MARVEL states with an RMS error of 0.056 cm$^{-1}$, a significant improvement to the 0.13 cm$^{-1}$ RMS error
obtained with the POKAZATEL (Polyansky et al., 2018) PES. At high temperatures, transitions between such high $J$ states
become significantly more prominent when compared to room temperature and hence this potential will offer improvements in
calculated line positions.

Combining our new surface with the CKAPTEN (Conway et al., 2018) dipole moment surface (DMS), we calculate a line
list which extends to 41 200 cm$^{-1}$, slightly beyond dissociation and includes transitions with $J_{\mathrm{max}} = 20$ possessing a minimum
intensity threshold of $10^{-32}$ cm molecule$^{-1}$. This line list is, however, not verified for transitions between 37 000 cm$^{-1}$ and
41 200 cm$^{-1}$ and basis set convergence issues may arise and influence line position accuracy.

This DMS has previously been verified through a significant number of comparisons against experimental and theoreti-
cal sources (Conway et al., 2020a, b) although not much is known in the near-ultraviolet. Comparisons of our new line list
against the POKAZATEL list indicate that there are relatively large differences in the visible and near-ultraviolet regions and
POKAZATEL underestimates the absorption. We show the change in potential is not the underlying cause of the discrepancies,
but rather the change in the DMS.

For wavelengths below 400 nm, the POKAZATEL absorption features drop almost systematically, which explains the under-
absorption observed at 363 nm (Lampel et al., 2017). The absorption calculated in our new list does not have this systematic
drop. Several experimental measurements in the 350 - 290 nm region have previously been performed (Du et al., 2013; Pei
et al., 2020; Wilson et al., 2016), although none agree with each other. Our calculations suggest the $5 \times 10^{-26}$ cm molecule$^{-1}$
upper limit on absorption of Wilson et al. is correct, while the other sources (Du et al., 2013; Pei et al., 2020) appear to over-
estimate cross sections by at least an order of magnitude. In particular, the absorption predicted by Du et al. or Pei et al. in the
near-ultraviolet would interfere with atmospheric retrievals in a manner which is simply not observed (Lampel et al., 2017).
Further experimental work on the near-ultravioloet absorption by water vapor is therefore required to resolve these issues.

Considering the improvements this new potential surface has to offer for high temperature spectra, future work is planned on
this. The potential energy surface is available in the supplementary material as a FORTRAN F90 file along with the calculated
line list assuming 100% abundance. This line list will form basis for the HITRAN2020 line list in the visible and UV where it
will be supplied with best available experimental data, especially for broadening. The calculated line list will also be added to
the ExoMol (Tennyson et al., 2016) website in the ExoMol format.

*Data availability.* The data to this article is provided in the supplementary material.

*Code availability.* The Fortran code for the potential energy surface is provided in the supplementary material.



*Competing interests.* The authors declare that they have no conflict of interest.

*Acknowledgements.* The author would like to thank Tibor Furtenbacher and Attila G. Császár for providing energy levels originating from a
provisional update to the MARVEL database.

*Financial support.* We thank the UK Natural Environment Research Council for funding under grant NE/T000767/1. Development of the HITRAN and HITEMP databases is supported through the NASA Aura and PDART grants NNX17AI78G and NNX16AG51G. SY and JT thank the STFC Project No. ST/R000476/1.





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
