# Peer review of "A semi-empirical potential energy surface and line list for $\mathrm{H_2}^{16}\mathrm{O}$ extending into the near-ultraviolet"

_Atmospheric Chemistry and Physics, 2020_

## Referee Comment (RC1) · Alain Campargue (Referee) · 28 Apr 2020

(Also provided as attached file) Report on A semi-empirical potential energy surface and line list for H216O extending into the near-ultraviolet by EK Conway et al

This work aims to contribute to the understanding of the strong discrepancy existing between some recent low resolution measurements of water absorption in the UV and calculations of the water absorption spectrum in this region. The paper is pleasant to read and the reported results, although lacking experimental validation, are very convincing and question seriously the origin of the much stronger water absorption measured by Pei et al and Du et al around 300 nm

[Figure]

Conway et al optimized a semi-empirical PES for H216O up to 37000 cm-1 and J= 20 and against empirical energy levels and generate a new line list up to 41200 cm-1. The obtained list is found to increase the UV absorption compared to the previous POKAZATEL list which was questioned by Lampel et al near 363 nm. The new calculations are now consistent with the results of Lampel et al. It is worth noting that according to the authors and in spite of the importance given in the paper to the improved PES, the differences in the DMS used for POKAZATEL and in the present work are mainly responsible of the increased UV absorption. Nevertheless, the obtained increased absorption is far to be sufficient to bring theory in accordance with the above mentioned experimental works: the resulting calculated absorption cross-sections remain between one and three orders smaller than the experimental values by Pei et al and Du et al. As underlined by the authors, new highly sensitive measurements in the region are highly suitable to validate their list.

Below a number of questions and suggestions: - When comparing energy levels calculated from different PES (eg Fig. 1) is it straightforward to identify the same energy levels in the different data sets using only the rigorous labels (J, parity, symmetry) in particular in the high energy range that you are considering. Could you give details about the adopted procedure to associate the levels. - Concerning this Fig. 1, it seems that deviations larger than 0.5 cm-1 was excluded below 25000 cm-1? Could you comment? Little is said in the text about this Fig. - Table 1 should be converted in a Fig and this long series of numbers (with rms values with 6 digits!) could be provided as Supplementary Material. On the other side, I am missing information: the authors refined their PES against J = 0, 2, 5, 10, 15 and 20, representing approximately 4 000 states while Table 1 applies to all their MARVEL levels, correct? Were some empirical energy levels excluded? On which criterion? What about bending levels? In the IUPAC-TG dataset of H216O, about 18500 levels were determined. Here the total numbers appearing in Table 1 are significantly lower (10500?) Could you explain? In principle all the IUPAC-TG levels (in fact even more with the recent new observations) should be considered. Could you mention/discuss the levels which were excluded? - The J = 0, 2,

5, 10, 15 and 20 levels were used to refine the PES. Does it mean that the rms values given in Table 1 for J = 0, 2, 5, 10, 15 and 20 correspond to the same set of levels as those included in the fit? - Line 285 : "For energy levels in J = 20, our new surface predicts MARVEL states with an RMS error of 0.056 cm−1, a significant improvement to the 0.13 cm−1 RMS error obtained with the POKAZATEL (Polyansky et al., 2018) PES." I am wondering to which extend this statement is informative: POKAZATEL was only refined to states in J = 0, 2 and 5 while the present PES use levels in in J = 0, 2, 5, 10, 15 and 20. The reader does not know if the quoted rms applies for the same set of levels, which ones were excluded. (Note that Line 49, the value of the POKAZATEL rms is as 0.118 cm-1). The considered set of MARVEL energy levels is unclear. Reference to a submitted paper Furtenbacher et al., 2020 is given. The full significance of the above sentence requires more precision - In the conclusion, the references attached to the MARVEL energy levels are (Császár et al., 2007; Furtenbacher and Császár, 2012) which are related to the MARVEL procedure and do not provide the used empirical levels. This "MARVEL washing" of huge experimental efforts should be avoided. In this context, probably Tennyson2013 is a better reference.

- Figure 2 should be improved: it seems that continuous lines were used for the plot while sticks or, better, dots should be used. Due to overlapping POKAZATEL CK-APTEN is not visible and there are many other issues. May be restrict the range to 20000-40000 cm-1 and plot only the envelopes of the different lists to allow to distinguish them. Several panels?

- Line 308-309 "This line list will form basis for the HITRAN2020 line list in the visible and UV ...'. I am wondering if, as a principle, such announce should not be validated by the HITRAN scientific committee. May be "This line list will be proposed for the HITRAN2020 line list in the visible and UV...

- Figure 3. I am surprised by the poor correlation between the 0.03 cm-1 and 1 nm resolution spectra. Of course, it could be due to the variation of the density of lines which makes the cross section so different compared to the envelope of the 0.03 cm-1

[Figure]

spectrum (for instance near 300 nm). Could you check and increase the sampling of the 1nm spectrum in order to have a smooth line instead of this ugly broken red line (by the way increase its width to make it more visible)

- I am surprised to find no mention and comparison to the high quality CRDS measurements of individual absorption lines near 25300 cm-1 by Dupré et al JCP 2005 doi.org/10.1063/1.2055247. To the best of my knowledge, this is the highest frequency measurements of absorption line intensities.

In conclusion, considering, the quality of the reported results obtained using state-of-the art theoretical calculations and the importance of the water absorption for a number of atmospheric applications, I recommend publication of this paper once the above comments and suggestions will have been addressed.

A. Campargue Tuesday, 28 April 2020

Please also note the supplement to this comment:
https://www.atmos-chem-phys-discuss.net/acp-2020-286/acp-2020-286-RC1-supplement.pdf

---

## Referee Comment (RC2) · Anonymous Referee #2 · 29 May 2020

The manuscript by Conway et al describes a very detailed work on developing a "new" (I would rather say improved or updated) semi-empirical potential energy surface and predicted list of nominally IR but also near IR, optical rovibrational transitions for H2O, with predictions extending all the way up into the near ultraviolet. As water is by far the strongest and most highly populated molecular absorber in the earth's atmosphere, such high quality predictions constitute a topic of undoubtedly high impact for any terrestrial IR, optical, or UV spectroscopic observations. Over the years, Tennyson's group has developed extremely high accuracy methods for solving the exact rovibrational eigenvalues and eigenfunctions for H2O and other small molecules with advanced Hamiltonian methods. Tennyson and his colleague Polyansky are undeniably experts in this field, which the Tennyson group has "plowed" with terrific vigor for many years. I have no doubt about the accuracy and correctness of this paper's scientific contents. Tennyson is extremely good at what he does, in part because he is smart and very good at maths, but also by virtue of having focused on this topic for multiple years if not decades. He has developed many excellent potentials for $H_2O$ in particular, each one representing a small yet still significant improvement on the preceding one, and predicting bound states and rovibrational transitions increasingly closer to the 41145 cm-1 dissociation limit. This has resulted in many excellent papers to his name, on a similar topic of improving an already excellent potential energy surface for $H_2O$ and making valuable and accurate predictions with it. The reason for this process continuing is not for testing against high level ab initio results (which are likely to be lagging well behind) but for the sheer importance of the resulting numbers. This paper (but more clearly the tables of linelists that he makes available to anyone through HITRAN or email request) will be values and read by any scientist studying spectroscopy of objects through the earth's atmosphere, which is dense with absorbing water vapor and enormous absorption path lengths. This paper in particular focuses on reliable predictions in the UV region, which is a super high order overtone event requiring accurate dipole moment functions, rovibrational wavefunctions, and Born-Oppenheimer corrections. The fact that these predictions indicate only weak absorption features in the UV is certainly to be expected – it is in the tireless quantitation of these absorption spectra that distinguishes Tennyson's contribution and craft.
* * *

---

## Referee Comment (RC3) · Anonymous Referee #3 · 5 Jun 2020

The manuscript 'A semi-empirical potential energy surface and line list for H216O extending into the near-ultraviolet' is well and concisely written and is suitable for publication on ACP, as it also contributed to bridging the gap between the theoretical spectroscopic calculations, which are often published elsewhere and a large number of observations of atmospheric trace gases and their implications which can be found on ACP, which often lack a thorough discussion of errors in used literature cross-sections and/or missing absorbers. This manuscript may help to remind the scientists working on different fields that the spectroscopic analysis of absorption spectra measured in the atmosphere is by far not perfect and various small, but often yet important effects need to be considered to be able to present reliable and precise measurements from

which then conclusions regarding atmospheric chemistry might be drawn.

The cited literature is extensive, well chosen and mostly quite representative for the current state of research. However, literature about water vapour retrievals from other (older) satellites such as GOME, GOME2 and SCIAMACHY comes a bit short, at least Wagner et al AMT 2013 and references therein could be mentioned.

Figure 3 could be extended by the upper limits on water vapour absorption cross-section values inferred from measurements by Lampel et al 2017 ACP, shown there in Table 4. The data from the publication which is reduced to a spectral resolution of 1nm could be sampled better, it seems to be quite coarse at the moment.
* * *

---

## Author Comment (AC1) · 14 Jul 2020

We thank the reviewer Alain Campargue for taking the time to read our article thoroughly and for providing a very constructive review. We have tried to address the comments to the best of our ability, and we believe the quality and clarity of the article has been improved by the comments. Below, we address each comment individually and highlight such changes in the article.

1. *When comparing energy levels calculated from different PES (eg Fig. 1) is it straightforward to identify the same energy levels in the different data sets using only the rigorous labels (J, parity, symmetry) in particular in the high energy range that you*

*are considering. Could you give details about the adopted procedure to associate the levels.*

The rigorous quantum numbers alone are not enough to match a calculated state to the correct corresponding state in MARVEL. We needed to supplement the rigorous quantum labels with energy differences, which is where it becomes difficult and very often non-trivial, particularly in the near-UV with the high density of states to identify the correct match. We added new J states only after the potential surface was optimized to the previous J states considered. This helped keep the energy differences low enough in the next J to make a match more accurately/reliably. For example, we would refine J=0, then pre-calculate J=2 states with this PES and identify the J=2 states. Next, refine J=0 and 2. Calculate J=5 states, then match the J=5 states. Then refine J=0, 2, 5, etc. We now explain this in the text.

2. *Concerning this Fig. 1, it seems that deviations larger than 0.5 $cm^{-1}$ was excluded below 25000 $cm^{-1}$? Could you comment? Little is said in the text about this Fig.*

For the Figure, there is indeed a 0.5 $cm^{-1}$ threshold for matching states below 26 000 $cm^{-1}$. To make this figure, and to facilitate an equal comparison, we calculated all states with the new PES, POKAZATEL PES and PES15K for 0<=J<=20. Next, the same algorithm was used to match all calculated states from each PES to a respective state in MARVEL. This helped to ensure the calculated energies would be matched via the same procedure and facilitate an equal comparison and accurate RMS comparison. The energy threshold was set to 0.5 $cm^{-1}$ for E<26000 $cm^{-1}$, and 1 $cm^{-1}$ for states above this limit. Clearly, our PES states do not really need the 0.5 $cm^{-1}$ cut off, but those states calculated with the POKAZATEL PES do, else if it was lower, less states would be matched from POKAZATEL. This is explained in the text.

3. *Table 1 should be converted in a Fig and this long series of numbers (with rms values with 6 digits!) could be provided as Supplementary Material.*

We agree that figures would be preferable to tables. We have added three subfigures to represent E<15000 cm$^{-1}$, E<=26000 cm$^{-1}$ and E<=37000 cm$^{-1}$. We have moved the tables to the supplementary material.

4. *On the other side, I am missing information: the authors refined their PES against J = 0, 2, 5, 10, 15 and 20, representing approximately 4 000 states while Table 1 applies to all their MARVEL levels, correct?*
We did indeed refine to approximately 4000 states, which were in J=0, 2, 5, 10, 15, 20. The table represents the residuals for states in MARVEL both refined to (J=0, 2, 5, 10, 15, 20) and not refined to (J=1, 3, 4, 6, 7, 8, 9, 11, 12, 13, 14, 16, 17, 18, 19). It is important that the PES has a 'smooth' behavior and can accurately predict the states between the Js that have not been fit. Hence, we calculated the residuals for those states in the un-refined J's to further promote its accuracy. The table has been moved to the supplementary.

5. *Were some empirical energy levels excluded? On which criterion? What about bending levels?*
Some states were indeed excluded from the refinement. States were removed if their residuals appeared to be abnormally higher than other J states in the same vibrational band which showed low average residuals. We expect residuals to be similar in the same vibrational band. The high bending states, i.e $\nu_2$>4 were difficult to fit and some were removed for the reason previously mentioned: abnormally high residuals for some states in the same band. Often, those bands with very few states were also outliers. Text has been added to the article to explain why and when states were removed.

6. *In the IUPAC-TG dataset of $H_2^{16}O$, about 18500 levels were determined. Here the total numbers appearing in Table 1 are significantly lower (10500?) Could you explain? In principle all the IUPAC-TG levels (in fact even more with the recent new observations) should be considered. Could you mention/discuss the levels which were excluded?*
I can indeed explain the 10526 in (now supplementary) Table 1. This number represents the number of states with energy <= 15000 cm$^{-1}$ (see top of table, above J=0

averaged results) and J<=20 that were matched with MARVEL using our calculated states, the remaining have higher energies or larger Js. One can see we compared with 14030 states with E<=26000 cm$^{-1}$ and J<=20. There are a total of 10651 states in the latest edition of MARVEL used here that have J<=20 and E<=15000 cm$^{-1}$. This means that 125 were not matched to at all: either quantum labels are different, and/or the energy deviations are greater than 0.5 cm$^{-1}$. But the full set of MARVEL was considered.

7. *The J = 0, 2, 5, 10, 15 and 20 levels were used to refine the PES. Does it mean that the rms values given in Table 1 for J = 0, 2, 5, 10, 15 and 20 correspond to the same set of levels as those included in the fit?*
Yes, the residuals in Table 1 (for this PES) reflect the fitted states for J=0, 2, 5, 10, 15, 20 (minus outliers). Table is in the supplementary now so no line has been added to the paper.

8. *Line 285 : "For energy levels in J = 20, our new surface predicts MARVEL states with an RMS error of 0.056 cm$^{-1}$, a significant improvement to the 0.13 cm$^{-1}$ RMS error obtained with the POKAZATEL (Polyansky et al., 2018) PES." I am wondering to which extend this statement is informative: POKAZATEL was only refined to states in J = 0, 2 and 5 while the present PES use levels in in J = 0, 2, 5, 10, 15 and 20. The reader does not know if the quoted rms applies for the same set of levels, which ones were excluded. (Note that Line 49, the value of the POKAZATEL rms is as 0.118 cm$^{-1}$).*
The 0.118 cm$^{-1}$ is the number quoted by Polyansky et al. that reflects their fitted RMS for MARVEL states in their J=0, 2, 5. This has been noted in the article. The 0.13 cm$^{-1}$ RMS and the 0.056 cm$^{-1}$ RMS indeed refer to the same states, i.e (supplementary) Table 1, E<15000 cm$^{-1}$, J=20 averaged energies. These large differences are due to the insufficient coverage of Js considered for the refinement of the POKAZATEL PES and show how our PES is significantly more suited toward the generation of high temperature spectra that necessitates the use of high J transitions for completeness.

9. *The considered set of MARVEL energy levels is unclear. Reference to a submitted*

*paper Furtenbacher et al., 2020 is given. The full significance of the above sentence requires more precision.*

We have added the respective reference: J. Phys. Chem. Ref. Data 49, (2020); 10.1063/5.0008253, which recently appeared online. This is the latest MARVEL 2020 data release.

10. *In the conclusion, the references attached to the MARVEL energy levels are (Császár et al., 2007; Furtenbacher and Császár, 2012) which are related to the MAR-VEL procedure and do not provide the used empirical levels. This "MARVEL washing" of huge experimental efforts should be avoided. In this context, probably Tennyson2013 is a better reference.*

We have replaced these references with the citation to the latest MARVEL2020 article here, as it just appeared online) (and where necessary in the article that references MARVEL energy level comparisons).

11. *Figure 2 should be improved: it seems that continuous lines were used for the plot while sticks or, better, dots should be used. Due to overlapping POKAZATEL CK-APTEN is not visible and there are many other issues. May be restrict the range to 20000-40000 cm$^{-1}$ and plot only the envelopes of the different lists to allow to distinguish them. Several panels?*

Figure 2 has been modified to use dots over joined lines. Also, a zoomed in region of 22500-40000 cm$^{-1}$ is included within the figure for clarity in the near-UV.

12. *Line 308-309 "This line list will form basis for the HITRAN2020 line list in the visible and UV . . .". I am wondering if, as a principle, such announce should not be validated by the HITRAN scientific committee. May be "This line list will be proposed for the HITRAN2020 line list in the visible and UV. . .*

Yes. Perhaps this was an ambitious assumption. We have addressed it.

13. *Figure 3. I am surprised by the poor correlation between the 0.03 cm$^{-1}$ and 1 nm resolution spectra. Of course, it could be due to the variation of the density of lines*

*which makes the cross section so different compared to the envelope of the 0.03 cm$^{-1}$ spectrum (for instance near 300 nm). Could you check and increase the sampling of the 1nm spectrum in order to have a smooth line instead of this ugly broken red line (by the way increase its width to make it more visible)*

Yes, we can see how the low 1 nm resolution can appear 'ugly'. We removed the 1 nm comparison and instead add a 0.2 nm resolution comparison to the upper limit from Lampel et al. (2017) deduced from observations at a 0.7 nm resolution. A 0.7 nm resolution spectrum is again very coarse, hence why we choose 0.2 nm: it maintains the band structure while still agreeing with their upper limit.

14. *I am surprised to find no mention and comparison to the high quality CRDS measurements of individual absorption lines near 25300 cm$^{-1}$ by Dupré et al JCP 2005 doi.org/10.1063/1.2055247. To the best of my knowledge, this is the highest frequency measurements of absorption line intensities.*

We have added a citation to the article of Dupré et al. JCP 2005 in our introduction, and in the conclusion we mention it will be proposed to be added in future releases.

---

## Author Comment (AC2) · 14 Jul 2020

We thank the referee for their kind comments about the article and our work in general. The report does not suggest any changes to the paper.
* * *

---

## Author Comment (AC3) · 14 Jul 2020

We thank the reviewer for the feedback on our theoretical article. Below we try to address the comments on an individual basis.

1. The cited literature is extensive, well chosen and mostly quite representative for the current state of research. However, literature about water vapour retrievals from other (older) satellites such as GOME, GOME2 and SCIAMACHY comes a bit short, at least Wagner et al AMT 2013 and references therein could be mentioned.

The recent work by Borger et al. AMT (2020) using TROPOMI products was missed by us on submission but this has been added. References to retrievals using data from

GOME, GOME-2 and SCIAMCHY have been added.

2. Figure 3 could be extended by the upper limits on water vapour absorption crosssection values inferred from measurements by Lampel et al 2017 ACP, shown there in Table 4.

We entirely missed this table from Lampel et al 2017 ACP but it is an important set of results to include in our study. We have added the upper limit using their 2nd order polynomial fits to Figure 4 with the other measurements. Their upper limit assumes a 0.7 nm resolution, which, like the 1nm resolution we initially used, is very coarse. We have used a 0.2 nm resolution instead, which preserves the structure and is still below their upper limit, which proves this upper limit is adhered to by our calculations. This is explained in the article.

3. The data from the publication which is reduced to a spectral resolution of 1nm could be sampled better, it seems to be quite coarse at the moment.

We addressed this above in comment (2). It is now 0.2 nm and the structure is well preserved.

**ACPD**